# DDX3X Syndrome Behavioral Manifestations with Particular Emphasis on Psycho-Pathological Symptoms—A Review

**DOI:** 10.3390/biomedicines11113046

**Published:** 2023-11-14

**Authors:** Urszula Stefaniak, Roksana Malak, Ada Kaczmarek, Włodzimierz Samborski, Ewa Mojs

**Affiliations:** 1Department of Clinical Psychology, Poznan University of Medical Sciences, 60-812 Poznan, Poland; ewamojs@ump.edu.pl; 2Department and Clinic of Rheumatology, Rehabilitation and Internal Medicine, Poznan University of Medical Sciences, 61-545 Poznan, Poland; rmalak@ump.edu.pl (R.M.); slewandowska@orsk.pl (W.S.); 3Faculty of Medicine, Poznan University of Medical Sciences, 61-701 Poznan, Poland; adakaczmarek.11@gmail.com

**Keywords:** DDX3X syndrome, rare disease, intellectual disability, intellectual and developmental disability, autism spectrum disorder, self-injurious behaviors, generalized anxiety disorder, speech delay

## Abstract

(1) Background: Identification of typical behavioral manifestations in patients with DEAD-Box Helicase 3 X-linked gene (*DDX3X*) variants plays a crucial role in accurately diagnosing and managing the syndrome. The objective of this paper was to carry out a review of medical and public databases and assess the behavioral features of the DDX3X syndrome (DDX3X), with a particular focus on psycho-pathological symptoms. (2) Methods: An extensive computerized search was conducted in various databases, including PubMed, Medline Complete, Science Direct, Scopus, and Web of Science. Specific keywords and Medical Subject Headings were used to ensure the inclusion of relevant studies. The Preferred Reporting Items for Systematic Reviews and Meta-Analyses (PRISMA) guidelines were applied to assess the methodological quality of the manuscripts. (3) Results: Only nine papers out of the 272 assessed met the inclusion criteria. These articles revealed various psycho-pathological manifestations in patients with the DDX3X syndrome. Intellectual disability (ID) or developmental disability (DD), speech delay, autism spectrum disorder (ASD), attention deficit hyperactivity disorder (ADHD), generalized anxiety disorder (GAD), self-injurious behaviors (SIBs), sensory symptoms and sleep disturbance were demonstrated to be the most common psycho-pathological behavior manifestations. (4) Conclusions: Patients with the DDX3X syndrome manifest a wide spectrum of psycho-pathological symptoms. A comprehensive investigation of these symptoms in patients is essential for early diagnosis and effective therapy.

## 1. Introduction

According to current estimates, there are over 10,000 rare diseases (RDs). And that number is rising [1]. While there is no universal definition of RDs, it is a concept closely related to the threshold of prevalence [2]. The X-linked rare neurodevelopmental disorder (RNDD) we would like to describe is the DDX3X syndrome. *DDX3X* is a part of the DEAD-box helicase family, located on p11.3–11.23 on the X chromosome. Unlike many X-linked genes, *DDX3X* escapes X-inactivation in females. It has a widespread expression across human tissues, and, based on its function in RNA metabolism, it plays a pivotal role in the regulation of gene expression, cell cycle control, viral replication, and innate immunity [3,4,5,6]. The DDX3X syndrome is caused by a spontaneous mutation within the *DDX3X* gene at conception [7,8,9]. This RD can be inherited; however, it does not happen frequently [4,7,10]. Analyzed data from multiple large cohort studies indicates that the *DDX3X* gene has the highest proportion of missense variants identified among other autosomal dominant de novo monogenic RNDDs [11].

*DDX3X* de novo variants cause developmental delays (DD) both in females, as first proven in a large cohort study in 2015, as well as in males [7], as confirmed later in 2018 [4,10,12]. The gene-based prevalence of de novo mutations was estimated in a 2022 paper by extrapolating data from the number of neurodevelopmental disorders (NDDs) cohort cases [11]. *DDX3X* was the second most frequently mutated gene in DD, accounting for 0.37% of all de novo variants; this gave a prevalence of DDX3-related NDDs with IDD of 3.6/100,000 individuals [11]. Based on Snijders et al.’s findings, mutations in *DDX3X* are one of the most common causes of ID, accounting for 1–3% of unexplained ID in women [7]. It was also shown that this rare genetic disease is not only associated with ID but also with neurological symptoms, motor delays, behavioral problems, cardiac dysfunction, as well as ophthalmic and gastrointestinal abnormalities [7,8,9,12,13,14,15]. Notably, *DDX3X* variants were linked with peculiar brain MRI abnormalities and brain tumors [3,16]. Nowadays, the population of patients with DDX3X accounts for 848 known cases across 54 countries. In total, 809 of those are females, and the remaining 39 are males [17]. Compared to other methods, it is crucial to carry out a careful review of cases together with Whole Exome Sequencing (WES) before a final DDX3X syndrome diagnosis due to the fact that *DDX3X* was reported previously as mimicking cerebral palsy or diagnosed as ASD [18,19].

Psycho-pathological symptoms refer to the observable behavioral, cognitive, emotional, or physiological manifestations that are indicative of mental or psychological disorders [20,21,22]. Knowledge of the psycho-pathological symptoms of the DDX3X syndrome is crucial for an accurate and timely diagnosis. Many patients affected by this syndrome exhibit a spectrum of symptoms ranging from ID to ASD. A thorough grasp of these behavioral patterns might yield valuable diagnostic markers and help healthcare practitioners identify and treat patients earlier. A more thorough understanding of the symptoms can also drive the development of targeted therapies, behavioral interventions, and supporting tools. It is significant to optimize the management of the DDX3X syndrome and enhance the overall patient outcome [15]. There have been some papers published on the behavioral symptoms of the DDX3X syndrome [7,15,16,23,24]. However, a systematic review encompassing studies focusing on the DDX3X syndrome’s psycho-pathological manifestations has not been conducted yet. Thus, the main purpose of the present systematic review was to determine the specific psycho-pathological features exhibited by patients with DDX3X and the distribution pattern of their prevalence.

The results section was split into smaller subsections, where we discuss every psycho-pathological manifestation that has been found in the literature.

## 2. Materials and Methods

### 2.1. Declaration and Protocol

A review was performed in accordance with the PRISMA 2020 Statement guidelines (Preferred Reporting Items for Systematic Reviews and Meta-Analyses). The PRISMA statement and its extensions are evidence-based minimum recommendations to encourage transparent and comprehensive reporting of systematic reviews. This evolving set of recommendations is designed to ensure that all the elements of this type of research are reported accurately and transparently. In other words, the PRISMA statement serves as a guide to help authors articulate best what has been conducted, what has been discovered, and, in the case of a review process, what they intend to do. Therefore, we have used this tool to assess the quality of the article [25] (see Figure 1).

### 2.2. Search Strategy

As part of our systematic review, we performed computer-based searches of the following medical and public databases: PubMed, Medline Complete, Science Direct, Scopus, and Web of Science. We used the following combinations of keywords and MeSH terms: “DDX3X syndrome”, “intellectual disability,” “neurodevelopmental disorder *,” autism spectrum disorder *,” and “*DDX3X*.” This paper was compiled due to the need to summarize results from a number of new and important manuscripts on the DDX3X syndrome and to discover more about the syndrome with a special emphasis on psycho-pathological manifestations. The literature published between 2015 (the year the syndrome was discovered) and 2022 was considered. This review incorporated the results of searches performed between 5 November 2022 and 11 December 2022.

### 2.3. Research Protocols and Inclusion and Exclusion Criteria

We performed the searches by defining the types of populations (participants) and the types of outcomes consistent with published papers that described the DDX3X syndrome manifestations. We formulated the following questions: “What are the specific psycho-pathological features exhibited by patients with the DDX3X syndrome?” and “How is the prevalence of psycho-pathological features distributed among patients with the DDX3X syndrome?”. The following inclusion criteria were defined in this review: (1) article type: case study, cohort study, research, and review; (2) date of publication: from 2015, justified by the fact that the first paper on the DDX3X syndrome was published in 2015; (3) population: studies conducted on human participants with a diagnosed DDX3X syndrome; (4) indication: manuscripts which investigate the psycho-pathological behaviors were selected for this review; and (5) language restrictions: only originally English language papers were taken into consideration. 

We applied the following exclusion criteria: (1) population: animal models; (2) indication: no identification of psycho-pathological symptoms; (3) language restrictions: language other than English.

### 2.4. Study Selection

A computerized search was conducted for papers that present developmental, psychiatric/psychological, adaptive, and behavioral symptoms associated with the DDX3X syndrome. Firstly, duplicates were removed from the 272 identified records. Secondly, each paper was screened by two independent reviewers. A total of 150 articles were excluded from the review after evaluating their titles and abstracts, as they were not aligned with the objectives of this review. A paper was considered pertinent if the researchers reached a consensus that it was related to the study questions. All discrepancies between the reviewers were discussed and resolved. Moreover, the references of 17 prospectively eligible relevant papers were read to check for any additional publications of interest. Subsequently, to provide a comprehensive assessment of the DDX3X syndrome, manuscripts describing fewer than 10 probands were assessed rigorously by the reviewers. In such cases, the reviewers checked carefully whether the manuscript in question covered broadly the psycho-pathological symptoms associated with DDX3X syndrome. Papers that included only a cursory assessment of the psycho-pathological manifestations of the DDX3X syndrome were excluded from the analysis as they were deemed insufficient by the reviewers. The results of the study selection are shown in Figure 1.

### 2.5. Data Extraction and Analysis

Two authors independently extracted pertinent information from each paper using a predefined and standardized Microsoft Excel 365 data form. In the event of any discrepancies between the researchers, discussions were held to achieve consensus. Data on the author’s (authors’) names, publication year, study design and important findings were documented meticulously. All the data are presented descriptively in the table (Table 1).

### 2.6. Data Synthesis

Due to the substantial methodological variations observed, particularly in measuring and reporting mental health and psycho-pathological outcomes, a meta-analysis of the gathered studies was not conducted. Instead, a narrative approach was employed to synthesize the findings, which were subsequently summarized and presented in tables. By adopting this approach, a comprehensive overview of the research outcomes was obtained despite the methodological heterogeneity among the included studies. The narrative synthesis and tabular summaries provided a concise and informative representation of the findings, enabling a nuanced understanding of the collected data.

## 3. Results

In total, 272 papers were found as a result of the literature search. In total, 172 of those were identified on PubMed, 32 on Medline Complete, 23 on Science Direct, 23 on Scopus, and 22 on Web of Science. A total of 105 papers were excluded because they were duplicates. The reviewers extracted 17 prospectively relevant articles from the remaining 167 screened records. Eventually, only nine full texts met the standards and were included in the review. One was a case study (Stefaniak et al.), one was a review (Levy et al.), one was a research paper (Tang et al.), and six were cohort studies (Snijders et al., Wang et al., Beal et al., Lennox et al., Ng-Cordell et al., and Dai et al.) [7,8,9,12,15,19,23,24,26]. Most of the studies were retrospective.

### 3.1. Included Papers

Nine papers that met the inclusion criteria of this review provide information about the relevant behavioral features of the DDX3X syndrome (Table 1).

### 3.2. Results of Data Extraction 

Data extraction results are presented in Table 2 and Figure 2. ID or DD, communication/speech delay, ASD or autistic-like behaviors, ADHD, GAD, SIBs, sensory symptoms, and sleep disturbance were the most common psycho-pathological behavior manifestations. Moreover, all of the nine papers that evaluated DDX3X concluded ID and/or DD and speech delay or communication problems as the most common features [7,8,9,12,15,19,23,24,26]. A summary of the psycho-pathological symptoms of the DDX3X syndrome is shown in Figure 2. A detailed description of the results is presented in the subsections below.

**Table 2 biomedicines-11-03046-t002:** Summary of clinical characteristics in patients with the DDX3X syndrome in the published literature.

	Number of Patients Tested in Different Research Groups/Centers							
Patients with Identified Behavior	Lennox [9] n= 107	Snijders Blok [7] n= 38	Wang [12] n= 28	Mount Sinai * [15,24] n= 24	Ng-Cordell [23] n= 23	Dai [26] n= 23	Beal [8] n= 6	Stefaniak [19] n= 1
**1.ID/DD**	n = 106	n = 38 (ID/DD)	n = 28 (ID/D)	n = 22	n = 23	n = 23	n = 3	X
**2. Communication / speech delay**	n = 38 ^2a^	n/a	n/a	n = 13	n = 21	n = 23 ^2b^	n = 4 ^2c^	1
**3. Autism spectrum disorder**	n = 22 ^3a^	n = 20 ^3b^	n = 6 ^3c^	n = 14	n = 15 ^3d^	n = 3	n = 2 ^3e^	1
**4. Attention deficit hyperactivity disorder**	n = 16 ^3a^	n/a	n/a	n = 9	n/a	n/a	n/a	x
**5. Generalized anxiety disorder**	n/a	n/a	n/a	n = 1	n = 16	n/a	n/a	x
**6. Self-injury**	n/a	n/a	n/a	n/a	n = 13	n/a	n/a	x
**7. Sensory symptoms**	n/a	n/a	n/a	n = 14	n = 10	n/a	n/a	n = 1
**8. Sleep disturbance**	n/a	n/a	n/a	n = 15	n = 13	n/a	n = 2	n = 1

n—number of patients. n/a—not applicable. ID/DD—Intellectual Disability/Developmental Delays. 2—2a—nonverbal, 2b—single words/minimally verbal, 2c—speech/motor delays. 3—3a—assessed via review of available medical records, 3b—behavioral problems including one or more of ASD/hyperactivity/aggression, 3c—ASD or other behavioral issue, 3d—autistic-like behaviors, 3e—ASD/hyperactivity/aggression. * The results of Tang et al.’s study and Levy et al.’s study were collectively referred to as the Mount Sinai cohort, as patients with the DDX3X syndrome mentioned in both papers were examined by a team at a single research center, the Seaver Autism Center at the Icahn School of Medicine at Mount Sinai. Microsoft Word 365 was used to generate Table 2.

**Figure 2 biomedicines-11-03046-f002:**
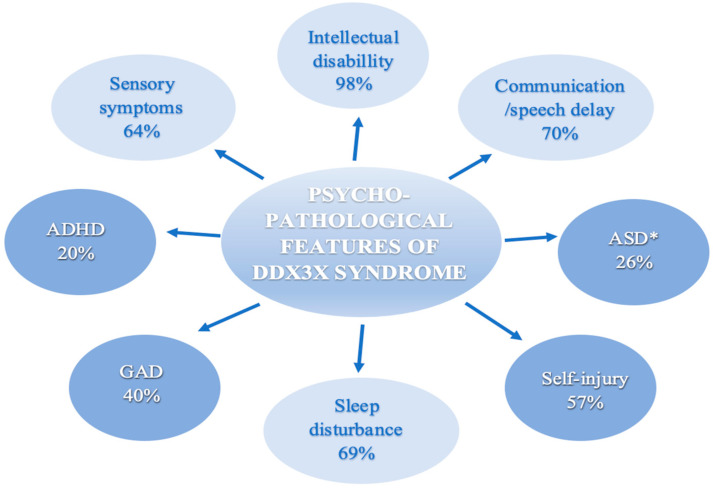
Summary of psycho-pathological features of the DDX3X syndrome. Abbreviations: ADHD = Attention-deficit and hyperactivity disorder. ASD = Autism spectrum disorder. GAD = Generalized anxiety disorder. When working with the published data, we avoided, as much as we could, double counting probands that were present in multiple studies. * Papers that did not provide specific information on ASD frequency were excluded from the calculations; Snijders Blok et al., Beal et al., Wang et al., Ng-Cordell et al. Figure 2 was generated using Microsoft Word 365, Version 2301 (Build 16026.20200).

### 3.3. Intellectual Disability

Eight out of the nine articles (Snijders et al., Wang et al., Beal et al., Lennox et al., Tang et al., Ng-Cordell et al., Dai et al. and Levy et al.) mentioned ID and/or DD as a DDX3X syndrome feature [7,8,9,12,15,23,24,26]. Snijders et al. identified 38 females with *DDX3X* de novo mutations that caused ID [7]. Wang et al. described an additional 28 out of 28 patients with DD or ID [12]. Patients between 1 and 33 years old were included. Wang et al. expanded the number of unique variants in *DDX3X* and described the additional 28 out of 28 patients with DD or ID. The ages of patients ranged from 1 to 47 years [12]. However, as proven by Beal et al., three out of six patients did not show any signs of ID [8]. Lennox et al. described that 106 out of 107 individuals met the criteria for ID [9]. In Tang et al.’s research, 12 out of 15 patients between 3 and 16 years old were diagnosed with ID [24]. Ng-Cordell et al. used the Developmental Behavior Checklist (DBC) as a scale to assess the emotional and behavioral characteristics of women with DDX3X. This tool was applied to measure T-scores for the severity of the ID. In total, 13 out of the 21 examined patients had scores above the clinical scale, which is a major concern [23]. The patient described in Stefaniak et al.’s case report seemed to have severe difficulties with social intelligence but did not manifest ID [19]. Dai et al. presented 23 out of 23 patients with ID [26]. The Mount Sinai cohort described by Levy et al. showed that intelligence levels ranged from average to severe, and adaptive behavior was similarly impaired. In most cases, the individuals demonstrated a delay across all the domains of adaptive functioning (including toilet training), with mean scores of 3 standard deviations below the population mean [15]. The total number of patients reported in the Mount Sinai cohort by Tang et al. and Levy et al. studies accounts for 22 out of 24 with ID [15,24].

ID is increasingly recognized and appears to be a universal feature among the DDX3X population. In total, 98% of individuals examined in the studies, which we included in our article, manifested an ID (Figure 2), where the majority of reported patients meet the criteria for ID ranging from mild to severe [7,9,12].

### 3.4. Communication Delay

Seven out of the nine articles (Beal et al., Lennox et al., Tang et al., Ng-Cordell et al., Stefaniak et al., Dai et al., and Levy et al.) mentioned communication or speech delay as a DDX3X syndrome feature (Table 2) [8,9,15,19,23,24,26]. Snijders et al. and Wang et al. did not present this feature as related to the DDX3X syndrome [7,12]. Beal et al. described speech or motor delays in four out of six patients, one of whom exhibited severe speech and language deficits despite having a mild to moderate ID diagnosis [8]. It was reported by Lennox et al. that past the age of 5, 52% of females with DDX3X were commonly nonverbal [9]. Tang et al. presented five individuals who were nonverbal out of a 15-patient cohort [24]. The average first-word milestone was achieved at 2.4 years, which is over a year later than expected. DDX3X individuals with ASD scored higher in all language assessments in comparison to those without ASD. Ten verbal participants’ expressive and receptive language skills were much below the age expectations [24]. Ng-Cordell et al. reported 21 out of 23 patients with communication or speech delay [23]. Problems in achieving communication milestones were reported as a relative weakness for 7 out of 23 individuals and a relative strength for 1; however, socialization skills were noted as an area of relative strength for 8 out of 23 participants and were not an area of weakness for any of the 23 participants [23]. In a recent case study, Stefaniak et al. described a 7-year-old female with a lack of ability to speak and a strong desire for social interaction at the same time [19]. That patient presented delays in expressive and receptive communication, but she was able to explore her environment, and she demonstrated social skills [19]. Language impairments were confirmed among 23 patients in the Chinese cohort by Dai et al., where 100% of participants could only say simple words and were minimally verbal [26]. The use of no more than four words and the ability to say single words was common in nearly 50% of participants. Two females over the age of five were non-verbal and could just follow simple commands [26]. In the Mount Sinai cohort, 13 out of 24 patients had speech delay with no words to few words. Language milestones were achieved with significant delays, with the first-word milestone at 31 months on average and at 48 months for phrase speech [15]. The unpublished Rush cohort mentioned in Levy et al., 2022 reports five patients with confirmed communication problems where single words were achieved at 2 and 3 years of age in three out of five patients, and two of them achieved phrase speech at the age of 4 [15].

There are no publications regarding the usage of sign language among patients as of today [27]. A recent Levy et al. review includes a mention that the use of augmentative and alternative communication (AAC) was an auspicious form of support to some DDX3X patients; however, the statistics were not given [15]. The authors of the review also refer to Rush’s unpublished cohort of five patients who had DD and ID simultaneously but presented well in terms of language and cognitive skills [15]. Levy et al. also noted an example of monozygotic female twins with the DDX3X syndrome who had discordant phenotypes; therefore, as is known from other genetic disorders, specific variants may not predict the clinical presentation of the patients [15]. One of the described twins had autism, was nonverbal, and presented aggressive behavior, and the other twin had the ability to use sentences but no presentation of ASD or aggressive behavior [15].

Referring to speech disorders, we reported the correlation between communication delay and the presence of *DDX3X,* and the results are statistically significant. ID was considered a universal feature of the DDX3X syndrome; nevertheless, communication impairments appear to be another striking characteristic of this neurodevelopmental disorder [7,11,15].

### 3.5. Autism Spectrum Disorder 

ASD characteristics include difficulty with verbal and nonverbal communication, repetitive behaviors or routines, difficulty with social interactions, and sensory sensitivities [28,29,30,31]. Thanks to a deeper understanding of the procedure for differentiating ASD from other disorders, clinicians have acquired new possibilities to facilitate an effective treatment of patients with neurodevelopmental conditions other than ASD, such as the DDX3X syndrome [9,15,19,24,26].

Referring to the Diagnostic and Statistical Manual of Mental Disorders (DSM-V), ASD affects mainly communication and behavior. Among patients with DDX3X syndrome, difficulties with friendship and sharing (from the Social Development Domain), verbal intonation, reciprocal conversation, and imaginative play (from the Language and Communication Domain), as well as auditory sensitivity (from the Interest and Behavior Domain) are noted most frequently [19,23]. Specific data on ASD were reported in Lennox et al., Tang et al., Stefaniak et al., Dai et al., and Levy et al. cohorts [9,15,19,24,26]. In a few papers, ASD symptoms were not extracted as separate but included along with aggression and hyperactivity in the category of behavioral problems or autistic-like behaviors (Table 2) [7,8,12,24]. In total, 40 participants out of the 153 examined in this regard (26%) had ASD symptoms (Figure 2). Lennox et al. discovered that individuals with *DDX3X* variants exhibited moderately higher levels of autism traits compared to the general population norms, as assessed by the Social Reactivity Scale (SR-II) and the Social Communication Questionnaire (SCQ) parent-report clinical tools [9]. According to Ng-Cordell et al.’s study, the Self-Absorbed subscale of the DBC, which primarily measures behaviors resembling those of autism, was frequently recognized as an issue of significant concern, with 15/21 participants indicating concern [23]. The items most frequently endorsed on this scale pertained to the organization of objects [23]. In a study conducted by Stefaniak et al., a new non-canonical splice-site variant of the *DDX3X* gene was identified, which exhibited symptoms resembling those of autism, such as repetitive tearing of paper, as well as some features that were not consistent with an ASD diagnosis, like spontaneous behaviors, curiosity, and a desire for social interaction despite being nonverbal [19]. The patient’s functioning improved following the implementation of inclusive education [19]. Behavioral interventions, such as a behavior analysis (ABA), were also helpful [19]. In the Chinese cohort, only 3 out of 23 patients had a confirmed ASD diagnosis [26]. However, a total of 13 participants were in the “above the risk” threshold for ASD according to the SCQ questionnaire administered, which should suggest the need for further diagnostic work in this direction [26]. The highest prevalence of ASD symptoms, found in 63% of patients, was observed in the Mount Sinai cohort, where all individuals underwent prospective and direct assessments using ADOS-2 (the Autism Diagnostic Observation Schedule-Second Edition), a validated observation tool, as well as ADI-R (Autism Diagnostic Interview-Revised), a lengthy and semi-structured parent interview [15].

Autistic spectrum disorder (ASD) symptoms may be a common psycho-pathological feature of the DDX3X syndrome that can manifest as a wide range of symptoms with varying degrees of severity [15,28,29].

### 3.6. Attention Deficit Hyperactivity Disorder

ADHD includes disruptive or maladaptive behaviors, such as impulsivity and hyperactivity. DDX3X patients can also experience these psychiatric or behavioral concerns [9,22,24]. Behavioral and/or pharmacological treatment and ongoing ADHD assessment are recommended. Consulting a child psychiatrist, developmental pediatrician, or neurologist is advisable [15].

Only two out of the nine manuscripts included ADHD as a feature of the DDX3X syndrome (Lennox et al. and Tang et al.) [9,24]. Snijders et al., Wang et al., Beal et al., Ng-Cordell et al., Stefaniak et al., Dai et al. and Levy et al. did not mention this disorder (Table 2) [7,8,12,15,19,23,26]. In Lennox et al., 16 out of 107 patients received an ADHD diagnosis [9]. In a recent research by Tang et al., 9 out of 22 patients demonstrated ADHD [24].

### 3.7. Abnormal Behavior

Two out of the nine articles (Ng-Cordell et al. and Levy et al.) mentioned GAD as a DDX3X syndrome feature (Table 2) [15,23]. Papers published previously, from 2015 to 2022, make no mention of it (Snijders et al., Wang et al., Beal et al., Lennox et al., Tang et al., Stefaniak et al., Dai et al.) [7,8,9,12,15,19,26]. Ng-Cordell et al. noted 16 out of 23 patients with signs of GAD [23]. Manifestations among the population of DDX3X patients include unusual fears of sounds and objects that were related to anxiety, including fire alarms, babies crying, hand dryers, sneezing, fireworks, onions, butterflies, and, for one participant, fear of animals with stripes, became an obsession. Shyness, social withdrawal, and worries about routine were also described as GAD [23]. In a review article by Levy et al., one patient with GAD is mentioned [15]. Levy presents an example of behavioral aggression when patients face valid fear or frustration and anxiety during separation from guardians and suggests patient-focused behavioral interventions and the possible use of pharmacology [15]. Moreover, it is recommended to verify mood and affect as DDX3X individuals are at risk for mood disorders [15]. GAD is a new psycho-pathological feature described in the literature, and it should be investigated further.

One of the nine articles (Ng-Cordell) presented SIBs [23]. Snijders et al., Wang et al., Beal et al., Lennox et al., Tang et al., Stefaniak et al., Dai et al. and Levy et al. did not mention this behavior as a DDX3X syndrome feature (Table 2) [7,8,9,12,15,19,24,26]. Ng-Cordell et al. were the first to determine SIBs as a statistically significant feature of the DDX3X syndrome that had not been reported before [23]. In contrast with other studies referred to above, Ng-Cordell et al. emphasized that more than half of the participants, 13 out of 23, presented a high incidence of SIBs, and it was significantly more common than in comparison to the ASD control group [23]. Females were pulling their hair, biting their hands or knees, banging their heads, and throwing themselves onto the floor in response to stressful situations [23]. To examine the kind of SIB behaviors, three DBC items were additionally checked: participants were banging their heads (11/23), picking or scratching their skin (13/23), and hitting or biting themselves (13/23). The author of the study mentioned in the discussion that anxieties and SIBs could be linked and that it can be a casual mechanism for SIBs in this particular RD [23]. As a new feature and area of serious concern, SIBs and their links with anxieties should be further investigated.

Three out of the nine articles (Tang et al., Ng-Cordell et al., and Stefaniak et al.) mentioned sensory symptoms as an important feature of the DDX3X syndrome [19,23,24]. There was no mention of this feature by Snijders et al., Wang et al., Beal et al., Lennox et al., Dai et al., or Levy et al. (Table 2) [7,8,9,12,15,26]. Atypical responses or unusual interest in sensory stimuli are behavioral reactions referred to as sensory features [32]. They, among others, consist of hyperreactivity, hyporeactivity, and overly focused sensory interests (referred to in the literature as sensory seeking) [32,33,34,35]. Sensory features were added as a new criterion in the Diagnostic and Statistical Manual of Mental Disorders (DSM-5) for the diagnosis of ASD, but they are also reported in other developmental disorders [32,35]. Tang et al. included 15 participants diagnosed with the DDX3X syndrome in the first prospective study, where definite sensory changes in the DDX3X syndrome were presented [24]. The researchers assessed sensory symptoms using the Short Sensory Profile (SSP), a career questionnaire, and the Sensory Assessment for Neurodevelopmental Disorders (SAND) questionnaire, which was administered by a clinical psychologist [24]. As part of the research conducted at the Seaver Autism Center, an adequate control group (*n* = 29) of typically developing individuals was assembled, and their z-score was calculated [24]. Sensory symptoms in all symptom domains and sensory modalities were more frequent in the DDX3X group (100%) compared to the control group. The results indicated that such sensory processing disorders as sensory seeking and hyporeactivity were more frequent than hyperreactive symptoms [24]. Furthermore, Tang et al. identified a more frequent occurrence of tactile hyporeactivity (e.g., high pain threshold) than visual and auditory hyporeactivity. In contrast, visual hyperreactivity was more frequent than tactile and auditory hyperreactivity [24]. A recent case report by Stefaniak et al., in the context of sensory processing disorders/dysfunctions, reported a dysregulation of temperature and tactile defensiveness, which made touching the patient extremely difficult [19]. For this reason, some therapeutic interventions could not be applied. Proprioceptive and vestibular stimulation was used as a therapy, and it made physical contact with the child possible and increased joint attention [19]. Similar conclusions about high levels of sensory hyporeactivity (80%) in patients with the DDX3X syndrome were noted in the Mount Sinai cohort [15]. The paper also showed high sensory-seeking behaviors (87%) [Levy]. Sensory symptoms described by Ng-Cordell et al. combined hypo and hypersensitivities, like high pain threshold and sensory-seeking behaviors, in <50% of patients [23].

Sensory deficits are a crucial component of the DDX3X syndrome. Usually, these are the earliest features noticed by parents and teachers, as they cause considerable disruption to emotional, social, and family life. Significantly, a high pain threshold can impact delayed reactions to dangerous stimuli, thus negatively affecting safety. Awareness of specific sensory symptoms and preferences can help parents and clinicians tailor the environment to the child’s needs [15,24].

In four out of the nine articles, sleep disturbance was mentioned as a DDX3X syndrome feature (Beal et al., Ng-Cordell et al., Stefaniak et al., and Levy et al.) [9,15,19,23]. Papers by Snijders et al., Wang et al., Lennox et al., Tang et al., and Dai et al. did not mention this feature (Table 2) [7,9,12,24,26]. Beal et al. listed 2 out of 6 patients with sleep disturbances [8]. Ng-Cordell et al. described 13 cases of patients who, during infancy or/and childhood, experienced hypersomnia or hyposomnia, early waking, difficulty falling asleep and maintaining sleep [23]. Sporadic sleep disturbances were also reported in a case review by Stefaniak et al. [19]. Levy et al. presented patients with sleep problems, issues associated with prolonged falling asleep and waking up in the middle of the night [15]. In the as-yet unpublished Rush paper mentioned by Levy et al., 3 out of 5 patients presented middle-of-the-night awakenings [15].

This statistic is significant in terms of providing a safe environment for patients at night if caregivers are asleep and is essential for reducing the severity of behavioral problems in the DDX3X population resulting from sleep deprivation and lack of attention during the day [15].

## 4. Discussion

The selection and criteria used in the present manuscript helped to identify nine papers that present the overall quality of the study [7,8,9,12,15,19,23,24,26]. Articles that were taken into consideration identified the following psycho-pathological manifestations among patients with the DDX3X syndrome: IDD, ASD, GAD, SIBs, sleep disturbances, ADHD, and sensory symptoms.

In Ng-Cordell et al.’s manuscript, the average age of patients is 12 (patients from the 3–22 age group) [19] (See Table 1). The oldest populations of patients were considered in Wang et al.’s research (1–47 years), then Snijders et al.’s (1–33 years), and Lennox et al.’s (1–24 years) [7,9,12]. A more in-depth examination of age-related variability in presentations and outcomes would strengthen the review’s insights and implications. However, the DDX3X syndrome is an extreme RD we still know very little about; this makes it impossible to discuss age-related variability in detail. We believe that through our systematic review, emphasis can be placed on exploring these issues in future clinical trials.

Interestingly, only Ng-Cordell et al. investigated SIBs and GAD among patients with the DDX3X syndrome in comparison with an ASD group of patients [23]; this suggests that in order to put the right management practices in place, more studies on self-injury among patients with the DDX3X syndrome, especially pediatric patients are needed. As reported by many parents, in contrast to SIBs, autistic-like behaviors, and anxiety in the DDX3X population, patients show a strong desire to be cooperative, caring, and friendly in social situations [23]. The positive social side of DDX3X patients has not yet been described or explored in the literature, apart from scant mentions of it in the medical interviews of parents and caregivers [19,23].

One study we came across investigated the correlation between genes and speech. That paper was not included in this review. Nonetheless, it is worth mentioning. Hildebrand et al.’s research proved that speech disorders might have a genetic origin [36]. In their study, for 11 out of 34 patients, childhood apraxia of speech (CAS) had a genetic background, and one of the genes identified as likely to cause speech disorder was *DDX3X* [36]. Speech disorders like CAS are serious social disorders, and understanding their genetic architecture could help identify accurate medical approaches. Furthermore, 9 out of 11 patients in that group did not inherit the mutation from their parents [36].

A full picture of behavioral manifestations in the DDX3X syndrome is not well-known or studied. Therefore, there is a need for more studies on larger groups and a control group in order to assess more widespread knowledge of this disease and its behavioral symptoms. Simultaneously, there is a significant requirement for WES screening in patients with unexplained ID and CAS, as *DDX3X* is a highly plausible pathogenic gene for both disabilities [7,36].

Accurate and early diagnosis may mean that children with DDX3X syndrome can benefit from proper therapy, educational process, and care. As a result of insufficient clinical data, not only formal diagnostic criteria for the DDX3X syndrome but also formal practice parameters for managing the care of patients with DDX3X have not been defined yet [15,27]. Patients are given an initial diagnosis of ASD, Rett Syndrome, Cerebral Palsy, or Toriello-Carey syndrome [17,37] and often have a diagnostic odyssey before they are referred for WES or next-generation sequencing testing (NGS) [15,18,19,38,39,40].

Exome sequencing (NGS or WES testing) means we have now discovered a larger group of genes that cause intellectual disability; this has an impact on differential diagnoses of other DDX3X genetic syndromes associated with ID [15,38,39,40]. For instance, *RSRC1* is associated with the features of global intellectual disability, developmental delay, conduct disorder, and hypotonia [38]. Variants in *RAC3* were linked to structural brain abnormalities and facial dysmorphia [39]. *ADGRL1* haploinsufficiency can lead to consistent developmental, neurological, and behavioral abnormalities [40]. That is why NGS and WES are especially important for patients who do not present strictly suggestive features for well-known developmental disorders. It could also shorten the diagnostic odyssey, as was shown in the early diagnosis of late-infantile neuronal ceroid lipofuscinosis type 2 (CLN2), and may enable the most accurate treatment for patients [41].

Even if patients meet the clinical features of such disorders as ASD, IDD, ADHD, GAD, SIBs, communication/speech delay, sensory symptoms, or CAS, a careful psychological review before a final diagnosis is crucial. It is also essential for psychologists with experience in assessing patients to make complex neurodevelopmental diagnoses of disorders, using measures appropriate for a given individual’s developmental level. For patients with each specific diagnosis, a specialized educational program based on standardized adaptive and academic testing is needed to support the individual’s unique learning profile. Today, a proper diagnosis results in a better chance of administering adequate treatment.

This review provides the most up-to-date knowledge on behavioral manifestations in patients with the DDX3X syndrome, with a particular focus on psycho-pathological symptoms based on the literature available in five databases. The number of studies on the newly discovered DDX3X syndrome has increased over the last few years. The insufficiency of available data, especially in terms of the small population of patients living in different locations around the world, suggests that a more specific study of these behavioral aspects and a wider research context is needed, particularly in the area of speech and relationship between CAS and the *DDX3X* mutation. The reported outcomes, such as the correlation between anxiety and SIB incidents for this group of patients, reflect the increased demand for clinical psychology support during childhood and adolescence.

## 5. Conclusions

Patients with the DDX3X syndrome manifest a broad spectrum of psycho-pathological symptoms. The key findings presented in this review are significant for diagnosing and managing the DDX3X syndrome. The findings can be used as a guide by professionals in order to provide an early diagnosis for children with DDX3X so that they, in turn, may benefit from appropriate therapy, education process, and care. A more detailed study of these behaviors is needed in future research.

## Figures and Tables

**Figure 1 biomedicines-11-03046-f001:**
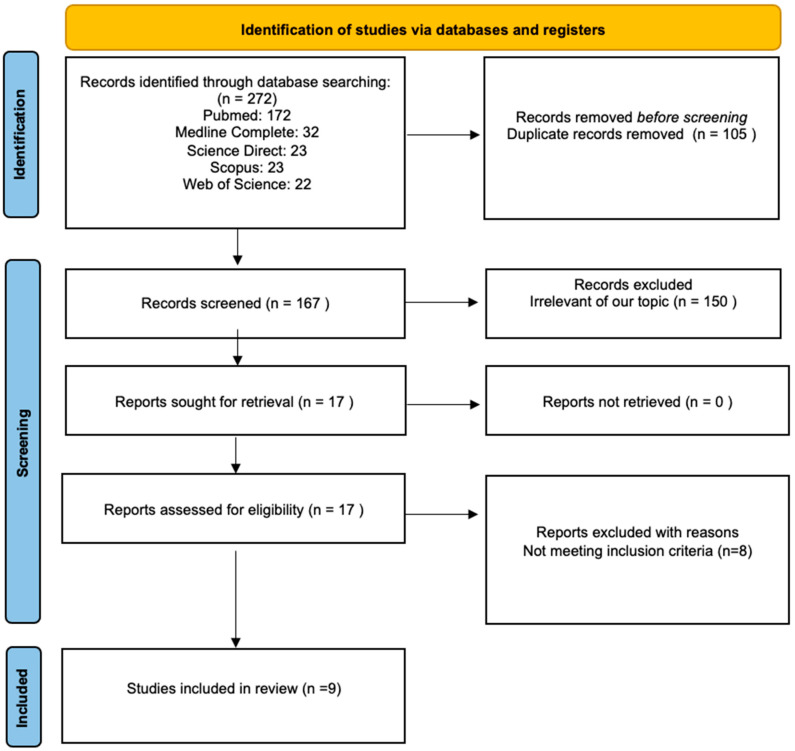
Flow chart of a review process according to the PRISMA guidelines [25].

**Table 1 biomedicines-11-03046-t001:** Characteristics of the included papers.

Study	Design	Year of Publication	Patient Age	Objective
Snijders et al., 2015 [7].	Cohort study	2015	1–33 years	Presentation of 37 unique deleterious de novo mutations in the *DDX3X* gene identified by whole exome sequencing.
Wang et al., 2018 [12].	Cohort study	2018	1–47 years	Description of DNA variants and phenotypes associated with DDX3X disorders.
Beal et al., 2019 [8].	Cohort study	2019	3–15 years	Characterization of the genotypic—phenotypic spectrum associated with heterogeneous DDX3X syndrome.
Lennox et al., 2020 [9].	Cohort study	2020	1–24 years	Elucidation of mechanisms by which pathogenic *DDX3X* variants disrupt brain development.
Tang et al., 2021 [24].	Research	2021	3–16 years	Expanding the knowledge of the neurobehavioral phenotype of the DDX3X syndrome in the first prospective study.
Ng- Cordell et al., 2021 [23].	Cohort study	2021	3–22 years	Comparison of social and emotional characteristics in patients with *DDX3X* variants to individuals with other monogenic causes of ID.
Stefaniak et al., 2022 [19].	Case report	2022	7 years	Introduction to symptoms of the musculoskeletal system and sensory integration processing disorders in the DDX3X syndrome.
Dai et al., 2022 [26].	Cohort study	2022	1–6 years	Identification of the clinical and genetic features of the DDX3X syndrome in Chinese patients.
Levy et al., 2023 [15].	Cohort study	2023	3–16 years	Presentation of clinical knowledge about the DDX3X syndrome and recommendations for clinical assessments.

## Data Availability

The authors confirm that the data which support the findings of this paper are openly available at https://doi.org/10.1016/j.ajhg.2015.07.004; https://doi.org/10.1097/MCD.0000000000000289; https://doi.org/10.1016/j.neuron.2020.01.042; https://doi.org/10.1002/acn3.622; https://doi.org/10.1016/j.pediatrneurol.2022.10.009; https://doi.org/10.3390/brainsci12030390; https://doi.org/10.1007/s10803-022-05527-w; https://doi.org/10.1186/s13229-021-00431-z; https://doi.org/10.3389/fnmol.2022.793001, reference numbers [9,10,11,12,15,19,23,24,26].

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
