# Peer review of "DDX3X Syndrome Behavioral Manifestations with Particular Emphasis on Psycho-Pathological Symptoms—A Review"

_biomedicines, 2023, doi:10.3390/biomedicines11113046_

Round 1

Reviewer 1 Report

Comments and Suggestions for Authors

The article is a review focusing on discerning the behavioral manifestations, particularly psychopathological symptoms, in patients with DDX3X syndrome, synthesized from nine selected studies out of 272. It scrutinizes articles published between 2015 and 2022, revealing diverse psychopathological manifestations such as IDD, ASD, GAD, SIBs, sleep disturbances, and ADHD. However, it does point out the existence of positive traits like cooperativeness, which are yet underexplored in the literature. The review emphasizes the variability in age range and the lack of studies exploring the correlation between the syndrome and speech disorders and their genetic origin.

The article needs to be revised and improved by addressing the following comments before it could be considered for publication:

·        The selection of only nine articles out of 272 appears overly restrictive, raising questions about the comprehensiveness of the study’s literature review. The exclusion of relevant articles may have narrowed the range of psychopathological symptoms evaluated and the resulting insights.

·        The text is vague on the specific inclusion criteria used for selecting the nine papers. This lack of transparency can mislead readers about the range and appropriateness of included studies.

·        The mixture of different types of studies, including case reports, cohort studies, and research studies, without adequate methodological appraisal or synthesis can lead to inconsistent and unreliable conclusions. More thorough synthesis and analysis of each article’s methodologies, findings, and limitations would provide readers with a clearer and more balanced understanding of the existing evidence.

·        The review seems to lack any meta-analytical approach or quantitative synthesis of the included studies’ results (such as using the funnel diagram), which limits the ability to draw robust conclusions from the existing literature.

·        The variability in age ranges across studies is noted but not adequately addressed or analyzed, reducing the review’s coherence and comprehensiveness. A more in-depth examination of age-related variability in presentations and outcomes would strengthen the review’s insights and implications.

·        The extent to which the identified psychopathological manifestations can be generalized to the broader DDX3X population is unclear, given the limited number and scope of the included studies and the lack of a control group.

·        The text mentions the positive social side of DDX3X patients but fails to delve deeper into this aspect. A more balanced view, addressing both positive and negative manifestations, would provide a more nuanced understanding of the syndrome.

·        The mention of potential genetic correlations between speech disorders and DDX3X is underdeveloped and raises questions about the relevance and implications of genetic factors in the manifestation and diagnosis of DDX3X syndrome. A more detailed exploration of genetic correlations would strengthen the article’s contribution to the understanding of DDX3X syndrome.

Summarising, while the article aims to provide an updated review of behavioral manifestations in DDX3X syndrome, focusing on psychopathological symptoms, several methodological limitations and conceptual weaknesses are evident. These include the restrictive and unclear inclusion criteria, the lack of methodological synthesis and analysis, and the limited exploration of the syndrome’s definition, manifestations, and genetic correlations. Addressing these issues would significantly enhance the review’s clarity, coherence, comprehensiveness, and contribution to the understanding of DDX3X syndrome.

Author Response

Dear Reviewer #1, 

Please find the answers attached in the file below. 

Kind regards, 

Urszula Stefaniak

Reviewer 2 Report

Comments and Suggestions for Authors

The authors present a systematic review of the pathophysiological symptoms recorded thus far for the rare developmental disorder known as DDX3X syndrome.  

The review was conducted carefully, according to PRISMA standards, and seems to be quite comprehensive.  A minor issue with the first Figure is that it appears that 22 papers identified from the Web of Science were left out, such that the sum of initial records identified is incorrect (250 instead of 272).  This omission also affects the next level of this figure:  142 instead of 167.

With respect to the overview of the results, I question the use of diagnostic terms (2, 3 and 4) in Table 2.  Unless there is evidence that a comorbid diagnosis can be alleged, it would be better to use descriptive terms (eg., attention deficit, hyperactivity instead of attention deficit/ hyperactivity disorder).  In this section, Figure 2 seems to be to be misplaced and should come after Table 2, not before.

With respect to specific areas of the findings, each section is extensively detailed.  This might be useful to practitioners in the field, but a somewhat more nuanced synthesis could be more informative to those who are not yet experienced in this specific disorder.  If one of the goals is to inform clinicians who might come in contact with suspicious cases, a few more general statements prior to providing the detailed breakdown would be helpful.

There is also some concern about including detailed discussion of the genetic heterogeneity of the disorder in multiple sections, such as that on cognitive impairment that includes many statements unrelated to cognition.  If the authors wish to highlight the issue of multiple de novo mutations (lines 223-226) this information should go elsewhere. Communication problems are also included under cognition; should perhaps either be separate or included in the title of this section.

Autism spectrum disorder:  Did the reports that mentioned this entity diagnose ASD, or are they referring to autistic symptoms, and if the latter, what symptoms are mentioned.  ASD is not a unitary entity, especially with current DSM standards, and if the goal is to improve the ability to identify DDX3X prior to WES, a more nuanced description of the symptoms exhibited by the known cases would be particularly relevant.  

This concern also applies to the section entitled Attention deficit hyperactivity disorder.

I fully understand the difficulty of describing behaviours that might be related to DDX3X particularly in light of the many different mutations involved, and believe that this communication is a solid attempt to categorize many of them, but also feel that a re-evaluation of the presentation would better facilitate the stated goal of helping other physicians to arrive at an early diagnosis that in turn would help patient outcome.

Comments on the Quality of English Language

While the English is ok, the entire document needs a review for formatting, typos, removal of internal editing, etc.

Author Response

Dear Reviewer #2, 

Please find answers attached in the file below. 

Yours Sincerely,

Urszula Stefaniak

Reviewer 3 Report

Comments and Suggestions for Authors

Dear Author, 

1. Clarity and Organization:

   - The introduction could be more explicit about the significance of studying behavioral manifestations in DDX3X syndrome for diagnosis and management. It might also help to provide a brief overview of the paper's structure.

2. Methods Section:

   - Specify the date range for the articles published between 2015 and 2022. For instance, you could clarify whether this is due to recent advances in research or for any other specific reason.

3. Methods Section:

   - In the methods section, it would be helpful to briefly explain the PRISMA guidelines and why they were chosen for assessing the methodological quality. Additionally, consider including the specific criteria used to assess the quality of the articles.

4. Results Section:

   - Provide a concise summary of the findings from the nine selected papers. This could include a brief description of the psychopathological manifestations observed in DDX3X patients.

5. Conclusion:

   - Consider adding a concluding paragraph that summarizes the key findings of the review and emphasizes their significance for diagnosing and managing DDX3X syndrome.

6. Citations and References:

   - Ensure consistent formatting of citations and references throughout the paper. Make sure they follow the required style guide (e.g., APA, MLA).

7. Grammar and Language:

   - Check for sentence structure and grammar issues throughout the paper to improve its overall readability.

8. Figures and Tables:

   - consider including relevant figures or tables to visually represent the key findings or statistical data from the review.

9. **Acknowledgments**:

If there are any acknowledgments to be made (e.g., funding sources or contributors), ensure they are included.

10. Abstract:

    - The abstract should provide a concise summary of the paper's objectives, methods, key findings, and implications. Make sure it accurately reflects the content of the paper.

11. Keywords:

    - Ensure that the keywords chosen for the paper are relevant to the content and help with discoverability in academic databases.

12. Acronyms and Abbreviations:

    - When introducing abbreviations such as "DDX3X," provide a full explanation of the acronym the first time it's used in the paper.

Comments on the Quality of English Language

Moderate editing of English language required

Author Response

Dear Reviewer #3, 

Please find answers attached in the file attached below. 

Yours Sincerely, 

Urszula Stefaniak

Round 2

Reviewer 1 Report

Comments and Suggestions for Authors

The authors have revised well. The manuscript is suitable for publication.

Author Response

Dear Reviewer #1, 

Thank you very much for your positive feedback on our manuscript.

Sincerely Yours, 

Urszula Stefaniak, Corresponding Author

Poznan University of Medical Sciences

Department of Clinical Psychology

Bukowska 70, 60-812 Poznan

email: [email protected]

Reviewer 3 Report

Comments and Suggestions for Authors

Dear Author,

Thanks for submitting your research manuscript entitled “DDX3X SYNDROME BEHAVIORAL MANIFESTATIONS WITH PARTICULAR EMPHASIS ON PSYCHO-PATHOLOGICAL SYMPTOMS- A REVIEW".

Note: Before giving my final comments, as well as the final revision of this manuscript,

Firstly, the author needs to address the following comments scientifically.
Major concerns:-
Please find out the following comments

This paper, which aims to assess the behavioral features of DDX3X syndrome with a focus on psychopathological symptoms, presents several significant issues that make it unsuitable for publication. Here are strong rejection reviewer comments:

1. Limited Scope and Significance: The paper focuses on a very narrow scope—examining the behavioral features of DDX3X syndrome with a focus on psychopathological symptoms. While understanding these features is important, the paper lacks broader context and significance. It does not explain why this syndrome is important, what its clinical implications are, or how the findings contribute to our understanding of the syndrome.

2. Inadequate Methodology: The paper's methodology and search strategy are insufficiently detailed. It fails to provide information about the specific search terms and criteria used to identify the relevant articles. Without this information, readers cannot assess the rigor of the search process, and the methodology lacks transparency.

3. Lack of Critical Evaluation: The paper does not critically evaluate the quality or limitations of the included studies. It is crucial to assess the methodological strengths and weaknesses of the selected articles to determine the reliability of the findings. Without this critical assessment, the paper's conclusions lack a strong evidentiary basis.

4. Data Presentation and Analysis: The paper does not provide a clear presentation of the data from the nine selected papers. It lacks statistical analysis, summaries of key findings, and comparisons among the studies. The reader is left with a superficial overview of the results without a deeper understanding of the psychopathological manifestations in DDX3X patients.

5. Lack of Discussion and Implications: The paper presents the results but does not discuss their clinical or scientific implications. It is essential to provide an interpretation of the findings and discuss how they may impact the diagnosis and management of DDX3X syndrome or contribute to the broader field of medical genetics.

6. Language and Clarity: The language and structure of the paper require improvement for clarity and readability. Some sentences are overly complex, and the flow of the paper is not well-organized. It is crucial to present the information in a clear and coherent manner.

7. Lack of Conclusive Insights: The paper fails to provide a clear and concise conclusion. It is essential to summarize the main findings and their implications for clinical practice and future research.

In summary, this paper lacks a clear research question, detailed methodology, critical evaluation of the literature, and meaningful discussion of the findings. It does not provide a significant contribution to the field and needs substantial revision and improvement before it can be considered for publication.

Comments on the Quality of English Language

 English very difficult to understand/incomprehensible

Author Response

Dear Reviewer #3, 

Thank you for your suggestions. Please the file attached below with our responses. 

Yours Sincerely, 

Urszula Stefaniak, Corresponding Author

Poznan University of Medical Sciences

Department of Clinical Psychology

Bukowska 70, 60-812 Poznan

email: [email protected]

Round 3

Reviewer 3 Report

Comments and Suggestions for Authors

Dear author,

After careful revision, manuscript revised successfully, and can be proceed further for publication.

Comments on the Quality of English Language

Minor editing of English language required

Author Response

Dear Reviever #3, 

We would like to thank you for your positive feedback on the manuscript. We have edited the manuscript to address your concerns regarding the use of language. The changes are highlighted in the text.

Yours Sincerely, 

Urszula Stefaniak, Corresponding Author

Poznan University of Medical Sciences

Department of Clinical Psychology

Bukowska 70, 60-812 Poznan

email: [email protected]